# Watermark-based Detection and Attribution of AI-Generated Image

## Abstract

Several companies–such as Google, Microsoft, and OpenAI–have deployed techniques to watermark AI-generated images to enable proactive detection. However, existing literature mainly focuses on *user-agnostic* detection. *Attribution* aims to further trace back the user who generated a detected AI-generated image. Despite its growing importance, attribution is largely unexplored. In this work, we aim to bridge this gap by providing the first systematic study on watermark-based, user-aware detection and attribution of AI-generated images. Specifically, we theoretically study the detection and attribution performance via rigorous probabilistic analysis. Moreover, we develop an efficient algorithm to select watermarks for the users to enhance attribution performance. Both our theoretical and empirical results show that watermark-based detection and attribution inherit the accuracy and (non-)robustness properties of the watermarking method.

## 1 Introduction

Generative AI (*GenAI*) can synthesize very realistic-looking images. Beyond its societal benefits, GenAI also raises ethical concerns. For instance, they can be misused to generate harmful images (Yang et al., 2024); they can be used to aid disinformation and propaganda campaigns by generating realistic-looking images (Dhaliwal, 2023); and people can falsely claim copyright ownership of images generated by them (Escalante-De Mattei, 2023).

Watermark-based detection and attribution of AI-generated images is a promising technique to mitigate these ethical concerns. For instance, several companies–such as Google, OpenAI, Stability AI, and Microsoft–have deployed such techniques to watermark their AI-generated images. Specifically, OpenAI inserts a visible watermark into the images generated by its DALL-E 2 (Ramesh et al., 2022); Google's SynthID (Gowal & Kohli, 2023) inserts an invisible watermark into images generated by its Imagen; Stability AI deploys a watermarking method in its Stable Diffusion (Rombach et al., 2022); and Microsoft watermarks all AI-generated images in Bing (Mehdi, 2023).

However, existing literature mainly focuses on *user-agnostic detection* of AI-generated images. In particular, the same watermark is inserted into all the images generated by a GenAI service; and an image is detected as generated by the GenAI service if a similar watermark can be decoded from it. *Attribution* aims to further trace back the registered user of the GenAI service who generated a given image.[1] Such attribution can aid the GenAI service provider or law enforcement in forensic analysis of cyber-crimes, such as disinformation and propaganda campaigns, that involve a given AI-generated image. Despite the growing importance of attribution, it is largely unexplored.

In this work, we bridge this gap by conducting the first systematic study on the *theory*, *algorithm*, and *evaluation* of watermark-based detection and attribution of AI-generated images. *Our work assumes an image watermarking method has been designed.* Our contribution is to study the theory and algorithm of leveraging this watermarking method for AI-generated image detection and attribution (illustrated in Figure 1). When a user registers in a GenAI service, a watermark (i.e., a bitstring) is selected for him/her and stored in a watermark database. When a user generates an image using the GenAI service, the user's watermark is embedded into the image using the *watermark encoder*. An image is detected as AI-generated if the watermark decoded from the image is similar enough to *at*

---

[1]Attribution could also refer to tracing back the GenAI service that generated a given image, which we discuss in Section L.

Figure 1: Illustration of *registration*, *generation*, and *detection & attribution* phases of watermark-based detection and attribution.

*least* one user's watermark in the watermark database. Moreover, the image is further attributed to the user whose watermark is the most similar to the decoded one.

We theoretically analyze the performance of watermark-based detection and attribution. Specifically, we define three key evaluation metrics: *true detection rate (TDR)*, *false detection rate (FDR)*, and *true attribution rate (TAR)*. We show that other relevant evaluation metrics can be derived from these three. Based on a formal quantification of a watermarking method's behavior, we derive lower bounds of *TDR* and *TAR*, and an upper bound of *FDR* no matter how the users' watermarks are selected.

Selecting watermarks for the users is a key component. We formulate a *watermark selection problem*, which aims to select a watermark for a new registered user via minimizing the maximum similarity between the selected watermark and the existing users' watermarks. We find that our watermark selection problem is equivalent to the well-known *farthest string problem* (Lanctot et al., 2003), which has been studied extensively in theoretical computer science. Thus, we adapt the *bounded search tree algorithm* (Gramm et al., 2003), a state-of-the-art solution to the farthest string problem, to solve our watermark selection problem.

We empirically evaluate our method for AI-generated images on three GenAI models, i.e., Stable Diffusion, Midjourney, and DALL-E 2. We use HiDDeN (Zhu et al., 2018), a deep-learning-based image watermarking method that is the basis for modern image watermarks. Our results show that detection and attribution are very accurate, i.e., *TDR/TAR* is close to 1 and *FDR* is close to 0, when AI-generated images are not post-processed; detection and attribution are still accurate when common post-processing, such as JPEG compression, Gaussian blur, and Brightness/Contrast, is applied to AI-generated images; and adversarial post-processing (Jiang et al., 2023) with a small number of queries to the detection API degrades the image quality substantially in order to evade detection/attribution. Moreover, we show our watermark selection algorithm outperforms baselines.

## 2 RELATED WORK

An image watermarking method typically consists of three components: *watermark*, *encoder*, and *decoder*. We consider a watermark $w$ to be a bitstring with $n$ bits. $E(C, w)$ means that encoder $E$ embeds $w$ into an image $C$, while $D(C')$ is the watermark decoded from a (watermarked or unwatermarked) image $C'$ by decoder $D$. Note that $E$ and $w$ can also be embedded into the parameters of a GenAI model such that its generated images are inherently watermarked with $w$ (Fernandez et al., 2023).

**Non-learning-based vs. learning-based:** Watermarking methods can be categorized into two groups based on the design of $E$ and $D$: *non-learning-based* and *learning-based*. Non-learning-based methods (Pereira & Pun, 2000; Bi et al., 2007; Wang, 2021; Wen et al., 2023) design $E$ and $D$ based on some hand-crafted heuristics, while learning-based methods (Zhu et al., 2018; Abdelnabi & Fritz, 2021; Luo et al., 2020; Wen & Aydore, 2019; Tancik et al., 2020; Fernandez et al., 2023) use neural networks as $E/D$ and automatically learn them using an image dataset. For instance, Tree-Ring (Wen et al., 2023) is a non-learning-based watermarking method, while HiDDeN (Zhu et al., 2018) is a learning-based method. Our theory and algorithm are applicable to both categories of watermarking methods as long as they use bitstring-based watermarks such as HiDDeN (Zhu et al., 2018), Stable Signature (Fernandez et al., 2023), StegaStamp (Tancik et al., 2020), and Smoothed HiDDeN (Jiang et al., 2024). We note that our results are not applicable to Tree-Ring, which employs a non-bitstring watermark. Since learning-based methods are more robust due to adversarial training (Zhu et al., 2018), we adopt a learning-based method in our experiments.

**Standard vs. adversarial training:** In learning-based methods, $E$ and $D$ are automatically learnt. Specifically, given an image $C$ and a random watermark $w$, the decoded watermark $D(E(C, w))$ for the watermarked image $E(C, w)$ should be similar to $w$, i.e., $D(E(C, w)) \approx w$. *Standard training* aims to jointly learn $E$ and $D$ such that $D(E(C, w))$ is similar to $w$ for an image dataset (Kandi et al., 2017). A watermarked image $E(C, w)$ may be post-processed, e.g., a watermarked image may be post-processed by JPEG compression during transmission on the Internet. Zhu et al. (2018) extended adversarial training (Goodfellow et al., 2015; Madry et al., 2018), a technique to train robust classifiers, to train watermarking encoder and decoder that are more robust against post-processing. Specifically, *adversarial training* aims to learn $E$ and $D$ such that $D(P(E(C, w)))$ is similar to $w$, where $P$ stands for a post-processing operation and $P(E(C, w))$ is a post-processed watermarked image. In each epoch of adversarial training, a $P$ is randomly sampled from a given set of them for each image in the image dataset.

**Robustness of watermarking:** We stress that building robust watermarking methods is *orthogonal* to our work and is still an ongoing effort. Non-learning-based methods (Pereira & Pun, 2000; Bi et al., 2007; Wang, 2021; Wen et al., 2023) are known to be non-robust to *common post-processing* such as JPEG compression (Zhu et al., 2018). Learning-based methods (Kandi et al., 2017; Zhu et al., 2018; Abdelnabi & Fritz, 2021; Luo et al., 2020; Wen & Aydore, 2019; Fernandez et al., 2023; Saberi et al., 2024) are more robust to such common post-processing because they can leverage adversarial training. For instance, common post-processing has to substantially decrease the quality of a watermarked image in order to remove the watermark (Luo et al., 2020; Wen & Aydore, 2019). *Adversarial post-processing* (Jiang et al., 2023; Lukas et al., 2024; Zhao et al., 2023; Saberi et al., 2024) strategically perturbs a watermarked image to remove the watermark. Learning-based image watermarking methods are not yet robust to adversarial post-processing in the white-box setting where an attacker has access to $D$. However, they have good robustness to adversarial post-processing when an attacker can only query the detection API for a small number of times in the black-box setting or does not have access to the detection API. In particular, adversarial post-processing substantially decreases the quality of a watermarked image in order to remove the watermark in such scenarios. To defend against adversarial post-processing, Jiang et al. (2024) proposed a framework to build certifiably robust image watermarks that cannot be removed when the $\ell_2$ norm of the added perturbation is bounded. *We acknowledge that our watermark-based detection and attribution inherit the watermarking method's (non-)robustness properties discussed above.*

## 3 PROBLEM FORMULATION

Suppose we are given a generative AI model, which is deployed as a GenAI service. A registered user sends a *prompt* (i.e., a text) to the GenAI service, which returns an AI-generated image to the user. In this work, we consider *detection* and *attribution* of AI-generated image. Detection aims to decide whether a given image was generated by the GenAI service or not; while attribution further traces back the user of the GenAI service who generated an image detected as AI-generated. Such attribution can aid the GenAI service provider or law enforcement in forensic analysis of cyber-crimes, e.g., disinformation or propaganda campaigns, that involve a given AI-generated image. We define the detection and attribution problems as follows:

**Definition 1** (Detection of AI-generated image). Given an image and a GenAI service, detection aims to infer whether the image was generated by the GenAI service or not.

**Definition 2** (Attribution of AI-generated image). Given an image, a GenAI service, and $s$ users $U = \{U_1, U_2, \cdots, U_s\}$ of the GenAI service, attribution aims to further infer which user used the GenAI service to generate the image after it is detected as AI-generated.

We note that the set of $s$ users $U$ in attribution could include all registered users of the GenAI service, in which $s$ may be very large. Alternatively, this set may consist of a smaller number of registered users if the GenAI service provider has some prior knowledge on its registered users. For instance, the GenAI service provider may exclude the registered users, who are verified offline as trusted, from the set $U$ to reduce its size. Moreover, malicious users may be identified by conventional network security solutions, such as IP addresses and behavior patterns (Yuan et al., 2019; Xu et al., 2021). How to construct the set of users $U$ in attribution is out of the scope of this work. Given any set $U$, our method aims to infer which user in $U$ may have generated a given image. We also note that

another relevant attribution problem is to trace back the GenAI service that generated a given image. Our method can also be used for such GenAI-service attribution, which we discuss in Section K.

## 4 DETECTION AND ATTRIBUTION

Figure 1 illustrates our watermark-based detection and attribution method. When a user registers in the GenAI service, the service provider selects a unique watermark for the user. We denote by $w_i$ the watermark selected for user $U_i$, where $i = 1, 2, \cdots, s$ is the user index. During generation, when a user $U_i$ sends a prompt to the GenAI service to generate an image, the provider uses the watermark encoder $E$ to embed watermark $w_i$ into the image. During detection and attribution, a watermark is decoded from a given image; the given image is detected as generated by the GenAI service if the decoded watermark is similar enough to at least one of the users' watermarks; and the given image is further attributed to the user whose watermark is the most similar to the decoded watermark after it is detected as AI-generated.

### 4.1 DETECTION

We use *bitwise accuracy* to measure similarity between two watermarks. Specifically, given any two watermarks $w$ and $w'$, their bitwise accuracy (denoted as $BA(w, w')$) is the fraction of matched bits in them: $BA(w, w') = \frac{1}{n} \sum_{k=1}^{n} \mathbb{I}(w[k] = w'[k])$, where $n$ is the watermark length, $w[k]$ is the $k$th bit of $w$, and $\mathbb{I}$ is the indicator function that has a value 1 if $w[k] = w'[k]$ and 0 otherwise. Given an image $C$, we use the decoder $D$ to decode a watermark $D(C)$ from it. We detect $C$ as AI-generated if there exists a user's watermark that is similar enough to $D(C)$, i.e., if the following is satisfied: $\max_{i \in \{1, 2, \cdots, s\}} BA(D(C), w_i) \geq \tau$, where $\tau > 0.5$ is the *detection threshold*.

### 4.2 ATTRIBUTION

Attribution is applied only after an image $C$ is detected as AI-generated. Intuitively, we attribute the image to the user whose watermark is the most similar to the decoded watermark $D(C)$. Formally, we attribute image $C$ to user $U_{i^*}$, where $i^*$ is as follows: $i^* = \arg \max_{i \in \{1, 2, \cdots, s\}} BA(D(C), w_i)$.

### 4.3 WATERMARK SELECTION

A key component of watermark-based detection and attribution is how to select watermarks for the users. Next, we first formulate watermark selection as an optimization problem, and then propose a method to approximately solve it.

#### 4.3.1 WATERMARK SELECTION PROBLEM

Intuitively, if two users have similar watermarks, then it is hard to distinguish between them for the attribution. In fact, our theoretical analysis in Section 5 shows that attribution performance is better if the maximum pairwise bitwise accuracy between the users' watermarks is smaller. Thus, we propose to select watermarks for the $s$ users to minimize their maximum pairwise bitwise accuracy. Formally, we formulate watermark selection as the following problem:

$$\min_{w_1, w_2, \cdots, w_s} \max_{i, j \in \{1, 2, \cdots, s\}, i \neq j} BA(w_i, w_j), \tag{1}$$

where $BA$ stands for bitwise accuracy between two watermarks. This optimization problem jointly optimizes the $s$ watermarks simultaneously. As a result, it is very challenging to solve the optimization problem because the GenAI service provider does not know the number of registered users (i.e., $s$) in advance. In practice, users register in the GenAI service at very different times. To address the challenge, we propose to select a watermark for a user at the time of his/her registration in the GenAI service. For the first user $U_1$, a random watermark is selected. Suppose watermarks for $s - 1$ users have been selected. Then, the $s$th user registers and the GenAI service provider selects a watermark $w_s$ whose maximum bitwise accuracy with the existing $s - 1$ watermarks is minimized. Formally, we formulate a *watermark selection problem* as follows:

$$\min_{w_s} \max_{i \in \{1, 2, \cdots, s-1\}} BA(w_i, w_s). \tag{2}$$

### 4.3.2 Solving the Problem

**NP-hardness:** Our watermark selection problem in Equation 2 turns out to be NP-hard. In particular, we can reduce the well-known NP-hard *farthest string problem* (Lanctot et al., 2003) to our watermark selection problem. The farthest string problem aims to find a string that is the farthest from a given set of strings. We can view a string as a watermark in our watermark selection problem, the given set of strings as the watermarks of the $s-1$ users, and the similarity metric between two strings as our bitwise accuracy. Then, we can reduce the farthest string problem to our watermark selection problem, which means that our watermark selection problem is also NP-hard. This NP-hardness implies that it is very challenging to develop an efficient exact solution for our watermark selection problem. We note that efficiency is important for watermark selection as a watermark is selected for a user at the time of registration. Therefore, we aim to develop an *efficient* algorithm that *approximately* solves the watermark selection problem.

**Random:** The most straightforward method to approximately solve the watermark selection problem in Equation 2 is to generate a $n$-bit bitstring uniformly at random as $w_s$. We denote this method as *Random*. The limitation of this method is that the selected watermark $w_s$ may be very similar to some existing watermarks, i.e., $\max_{i \in \{1,2,\cdots,s-1\}} BA(w_i, w_s)$ is large, making attribution less accurate, as shown in our experiments.

**Decision problem:** To develop an efficient algorithm to approximately solve our watermark selection problem, we first define its *decision problem*. Specifically, given the maximum number of matched bits between $w_s$ and the existing $s-1$ watermarks as $m$, the decision problem aims to find such a $w_s$ if there exists one and return *NotExist* otherwise. Formally, the decision problem is to find any watermark $w_s$ in the following set if the set is nonempty: $w_s \in \{w | \max_{i \in \{1,2,\cdots,s-1\}} BA(w_i, w) \leq m/n\}$, where $n$ is the watermark length. Next, we discuss how to solve the decision problem and then turn the algorithm to solve our watermark selection problem.

**Approximate bounded search tree algorithm (A-BSTA):** Our A-BSTA is an adapted version of the *bounded search tree algorithm (BSTA)*, the state-of-the-art *exact* algorithm to solve the decision problem version of the farthest string problem. The details of BSTA can be found in Appendix A. Our A-BSTA makes two adaptions of BSTA. First, we constrain the recursion depth $d$ to be a constant (e.g., 8 in our experiments) instead of $m$, which makes the algorithm approximate but improves the efficiency substantially. Second, instead of initializing $w_s$ as $\neg w_1$, we initialize $w_s$ as an uniformly random watermark. As our experiments in Table 2 in Appendix show, our initialization further improves the performance of A-BSTA. This is because a random initialization is more likely to have small bitwise accuracy with all existing watermarks. Note that A-BSTA returns *NotExist* if it cannot find a solution $w_s$ to the decision problem.

**Solving our watermark selection problem:** Given an algorithm (e.g., A-BSTA) to solve the decision problem, we turn it as a solution to the watermark selection problem. Our idea is to start from a small $m$, and then solve the decision problem. If we cannot find a watermark $w_s$ for the given $m$, we increase it by 1 and solve the decision problem again. We repeat this process until finding a watermark $w_s$. Note that we start from $m = \max_{i \in \{1,2,\cdots,s-2\}} n \cdot BA(w_i, w_{s-1})$, i.e., the maximum number of matched bits between $w_{s-1}$ and the other $s-2$ watermarks. This is because an $m$ smaller than this value is unlikely to produce a watermark $w_s$ as it failed to do so when selecting $w_{s-1}$. Algorithm 3 in Appendix shows our method.

## 5 Theoretical Analysis

We first formally define three key metrics to evaluate the performance of detection and attribution. Then, we theoretically analyze the evaluation metrics. All our proofs are shown in Appendix.

**Image distributions:** We denote the $s$ users' watermarks as a set $W = \{w_1, w_2, \cdots, w_s\}$. When a user $U_i$ generates an image via the GenAI service, the service provider uses the encoder $E$ to embed the watermark $w_i$ into the image. We denote by $\mathcal{P}_i$ the probability distribution of watermarked images generated by $U_i$. Note that two users $U_i$ and $U_j$ may have different AI-generated, watermarked image distributions $\mathcal{P}_i$ and $\mathcal{P}_j$. This is because two users have different watermarks and they may be interested in generating different types of images. Moreover, we denote by $\mathcal{Q}$ the probability distribution of non-AI-generated images.

## 5.1 Evaluation Metrics

**(User-dependent) True Detection Rate (TDR):** *TDR* is the probability that an AI-generated image is correctly detected. Note that different users may have different AI-generated image distributions. Therefore, *TDR* depends on users. We denote by $TDR_i$ the true detection rate for the watermarked images generated by user $U_i$, i.e., $TDR_i$ is the probability that an image $C$ sampled from $\mathcal{P}_i$ uniformly at random is correctly detected as AI-generated. Formally, we have:

$$TDR_i = \Pr_{C \sim \mathcal{P}_i}(\max_{j \in \{1,2,\cdots,s\}} BA(D(C), w_j) \geq \tau), \tag{3}$$

where the notation $\sim$ indicates an image is sampled from a distribution uniformly at random.

**False Detection Rate (FDR):** *FDR* is the probability that an image $C$ sampled from the non-AI-generated image distribution $\mathcal{Q}$ uniformly at random is detected as AI-generated. Note that *FDR* does not depend on users. Formally, we have:

$$FDR = \Pr_{C \sim \mathcal{Q}}(\max_{j \in \{1,2,\cdots,s\}} BA(D(C), w_j) \geq \tau). \tag{4}$$

**(User-dependent) True Attribution Rate (TAR):** *TAR* is the probability that an AI-generated image is correctly attributed to the user that generated the image. Like *TDR*, *TAR* also depends on users. We denote by $TAR_i$ the true attribution rate for watermarked images generated by user $U_i$, i.e., $TAR_i$ is the probability that an image sampled from $\mathcal{P}_i$ uniformly at random is correctly attributed to user $U_i$. Formally, we have:

$$TAR_i = \Pr_{C \sim \mathcal{P}_i}(\max_{j \in \{1,2,\cdots,s\}} BA(D(C), w_j) \geq \tau \tag{5}$$
$$\wedge BA(D(C), w_i) > \max_{j \in \{1,2,\cdots,s\}/\{i\}} BA(D(C), w_j)),$$

where the first term $\max_{j \in \{1,2,\cdots,s\}} BA(D(C), w_j) \geq \tau$ means that $C$ is detected as AI-generated, and the second term $BA(D(C), w_i) > \max_{j \in \{1,2,\cdots,s\}/\{i\}} BA(D(C), w_j)$ means that $C$ is attributed to user $U_i$. Note that we have the first term because attribution is only applied after detecting an image as AI-generated.

**Other metrics:** In Appendix B, we show other relevant metrics can be derived from $TDR_i$, *FDR*, and $TAR_i$.

## 5.2 Formal Quantification of Watermarking

Intuitively, to theoretically analyze the detection and attribution performance (i.e., $TDR_i$, *FDR*, and $TAR_i$), we need a formal quantification of a watermarking method's behavior at decoding watermarks in AI-generated and non-AI-generated images. Towards this end, we formally define $\beta$-*accurate watermarking* and $\gamma$-*random watermarking*, the details of which are in Appendix C.

$\beta$-accurate watermarking is used to characterize the accuracy of the watermarking method at encoding/decoding a watermark in an AI-generated image. In particular, the watermarking method is more accurate when $\beta$ is closer to 1. $\gamma$-random watermarking characterizes the behavior of the watermarking method for non-AI-generated images. In particular, the decoded watermark for a non-AI-generated (i.e., non-watermarked) image is close to a uniformly random watermark, where $\gamma$ quantifies the difference between them. The watermarking method is more random for non-AI-generated images if $\gamma$ is closer to 0.

**User-dependent $\beta_i$:** Since the users' AI-generated images may have different distributions $\mathcal{P}_i$, the same watermarking method may have different $\beta$ for different users. To capture this phenomena, we consider the watermarking method is $\beta_i$-accurate for user $U_i$'s AI-generated images embedded with watermark $w_i$. Note that the same $\gamma$ is used across different users since it is used to characterize the behavior of the watermarking method for non-AI-generated images, which is user-independent.

**Incorporating post-processing:** Our $\beta$-accurate and $\gamma$-random watermarking can also incorporate post-processing (e.g., JPEG compression) that an attacker may apply to AI-generated or non-AI-generated images. In particular, we can replace $D(C)$ as $D(P(C))$ in definitions, where $P$ stands for post-processing of the image $C$. When the AI-generated image is post-processed, the watermarking method may become less accurate, i.e., $\beta$ may decrease. The parameters $\beta$ and $\gamma$ can be estimated using a set of AI-generated and non-AI-generated images, as shown in our experiments.

## 5.3 Detection Performance

**Theorem 1** (Lower bound of *TDR$_i$*). *Suppose we are given $s$ users with any $s$ watermarks $W = \{w_1, w_2, \cdots, w_s\}$. When the watermarking method is $\beta_i$-accurate for user $U_i$, we have a lower bound of TDR$_i$:*

$$TDR_i \geq Pr(n_i \geq \tau n) + Pr(n_i \leq n - \tau n - \underline{\alpha_i} n), \tag{6}$$

*where $0.5 < \tau < \beta_i$, $\underline{\alpha_i} = \min_{j \in \{1,2,\cdots,s\}/\{i\}} BA(w_i, w_j)$, and $n_i \sim B(n, \beta_i)$ (binomial distribution).*

**Corollary 1.** *When the watermarking is more accurate, i.e., $\beta_i$ is closer to 1, the lower bound of TDR$_i$ is larger.*

**Theorem 2** (Upper bound of *FDR*). *Suppose we are given $s$ users with $s$ watermarks $W = \{w_1, w_2, \cdots, w_s\}$ and watermark $w_1$ is selected uniformly at random. We have an upper bound of FDR as follows:*

$$FDR \leq Pr(n_1 \geq \tau n) + Pr(n_1 \leq n - \tau n + \overline{\alpha_1} n), \tag{7}$$

*where $\overline{\alpha_1} = \max_{j \in \{2,3,\cdots,s\}} BA(w_1, w_j)$ and $n_1 \sim B(n, 0.5)$.*

Note that the upper bound of *FDR* in Theorem 2 does not depend on $\gamma$-random watermarking since we consider $w_1$ is picked uniformly at random. However, we found such upper bound is loose. This is because the second term of the upper bound considers the worst-case scenario of the $s$ watermarks. The next theorem shows that when the $s$ watermarks are constrained, in particular selected independently, we can derive a tighter upper bound of *FDR*.

**Theorem 3** (Alternative upper bound of *FDR*). *Suppose we are given $s$ users with $s$ watermarks $W = \{w_1, w_2, \cdots, w_s\}$ selected independently. When the watermarking method is $\gamma$-random for non-AI-generated images, we have an upper bound of FDR as follows:*

$$FDR \leq 1 - Pr(n' < \tau n)^s, \tag{8}$$

*where $n' \sim B(n, 0.5 + \gamma)$.*

**Corollary 2.** *When the watermarking method is more random for non-AI-generated images, i.e., $\gamma$ is closer to 0, the upper bound of FDR is smaller.*

**Impact of $s$ on the bounds:** Intuitively, when there are more users, i.e., $s$ is larger, it is more likely to have at least one user whose watermark has a bitwise accuracy with the decoded watermark $D(C)$ that is no smaller than $\tau$. As a result, both *TDR$_i$* and *FDR* may increase as $s$ increases, i.e., $s$ controls a trade-off between *TDR$_i$* and *FDR*. Our theoretical results align with this intuition. On one hand, Theorem 1 shows that the lower bound of *TDR$_i$* is larger when $s$ is larger. In particular, when $s$ increases, the parameter $\underline{\alpha_i}$ may become smaller. Thus, the second term of the lower bound increases, leading to a larger lower bound of *TDR$_i$*. On the other hand, the upper bound of *FDR* in both Theorem 2 and Theorem 3 increases as $s$ increases. In particular, in Theorem 2, $\overline{\alpha_1}$ becomes larger when $s$ increases, leading to a larger second term of the upper bound.

**User-agnostic vs. user-aware detection:** Existing watermark-based detection is user-agnostic, i.e., it does not distinguish between different users when embedding a watermark into an AI-generated image. The first term of the lower bound in our Theorem 1 is a lower bound of *TDR* for user-agnostic detection; the first term of the upper bound in our Theorem 2 is an upper bound of *FDR* for user-agnostic detection; and the upper bound with $s = 1$ in our Theorem 3 is an alternative upper bound of *FDR* for user-agnostic detection. Compared to user-agnostic detection, user-aware detection achieves larger *TDR* but also larger *FDR*.

## 5.4 Attribution Performance

**Theorem 4** (Lower bound of *TAR$_i$*). *Suppose we are given $s$ users with any $s$ watermarks $W = \{w_1, w_2, \cdots, w_s\}$. When the watermarking method is $\beta_i$-accurate for user $U_i$, we have a lower bound of TAR$_i$ as follows:*

$$TAR_i \geq Pr(n_i \geq \max\{\lfloor \frac{1 + \overline{\alpha_i}}{2} n \rfloor + 1, \tau n\}), \tag{9}$$

*where $\overline{\alpha_i} = \max_{j \in \{1,2,\cdots,s\}/\{i\}} BA(w_i, w_j)$ and $n_i \sim B(n, \beta_i)$.*

Our Theorem 4 shows that the lower bound of $TAR_i$ is larger when $\beta_i$ is closer to 1, i.e., attribution performance is better when the watermarking method is more accurate. Moreover, the lower bound is larger when $\overline{\alpha_i}$ is smaller because it is easier to distinguish between users. This is a theoretical motivation on why our watermark selection problem aims to select watermarks for the users such that they have small pairwise bitwise accuracy.

**Detection implies attribution:** When $\tau > \frac{1+\overline{\alpha_i}}{2}$, the lower bound of $TAR_i$ in Theorem 4 becomes $TAR_i \geq \Pr(n_i \geq \tau n)$. The second term of the lower bound of $TDR_i$ in Theorem 1 is usually much smaller than the first term. In other words, the lower bound of $TDR_i$ is also roughly $\Pr(n_i \geq \tau n)$. Therefore, when $\tau$ is large enough (i.e., $> \frac{1+\overline{\alpha_i}}{2}$), $TDR_i$ and $TAR_i$ are very close, which is also confirmed in our experiments. This result indicates that once an AI-generated image is correctly detected, it would also be correctly attributed.

# 6 EXPERIMENTS

## 6.1 EXPERIMENTAL SETUP

**Datasets:** We consider both AI-generated and non-AI-generated images. For AI-generated, we use three public datasets (Wang et al., 2023; Turc & Nemade, 2022; Images, 2023) generated respectively by Stable Diffusion, Midjourney, and DALL-E 2. Following HiDDeN (Zhu et al., 2018), for each dataset, we sample 10,000 images for training watermark encoders and decoders; and we sample 1,000 images for testing. For non-AI-generated, we combine the images in COCO (Lin et al., 2014), ImageNet (Deng et al., 2009), and Conceptual Caption (Sharma et al., 2018), and sample 1,000 images from the combined set uniformly at random as our non-AI-generated dataset. We scale the image size in all datasets to be $128 \times 128$.

**Watermarking method:** We use the learning-based method HiDDeN (Zhu et al., 2018) because it is the basis of modern image watermarks like Stable Signature (Fernandez et al., 2023), StegaStamp (Tancik et al., 2020), and Smoothed HiDDeN (Jiang et al., 2024). Unless otherwise mentioned, we use standard training with the default parameter settings in the publicly available code. For each GenAI model, we train a watermark encoder/decoder using the corresponding AI-generated image training set and evaluate performance on the testing set.

**Evaluation metrics:** We use *TDR*, *FDR*, and *TAR*. *FDR* is the fraction of the 1,000 non-AI-generated images that are falsely detected as AI-generated. For each user $U_i$, we embed its watermark into 100 images randomly sampled from a testing AI-generated image dataset; and then we calculate the $TDR_i$ and $TAR_i$ for the user. In most of our experiments, we report the *average TDR* and *average TAR*, which respectively are the average $TDR_i$ and $TAR_i$ among the $s$ users. However, average *TDR* and average *TAR* cannot reflect the detection/attribution performance for the worst-case users, i.e., some users may have quite small $TDR_i/TAR_i$, but the average *TDR/TAR* may still be very large. Therefore, we further consider the 1% users (at least 1 user) with the smallest $TDR_i$ (or $TAR_i$) and report their average *TDR* (or *TAR*), which we call *worst 1% TDR* (or *worst 1% TAR*).

**Parameter settings:** By default, we set $s = 100,000$ (due to limited computation resource), $n = 64$, and $\tau = 0.9$. We also explore $s = 1,000,000$. Unless otherwise mentioned, we show results for the Stable Diffusion dataset.

## 6.2 DETECTION AND ATTRIBUTION RESULTS

**Without post-processing:** We first show results when the AI-generated, watermarked images are not post-processed. For each GenAI model, we compute the $TDR_i/TAR_i$ of each user and the *FDR*. The *FDR*s for the three GenAI models are nearly 0. Then, we rank the users' $TAR_i$ (or $TDR_i$) in a non-descending order. Figure 2a shows the ranked $TAR_i$ of the 100,000 users for the three GenAI models. Note that the curve of $TDR_i$ overlaps with that of $TAR_i$ for a GenAI model and thus is omitted in the figure for simplicity. $TDR_i$ and $TAR_i$ overlap because $\tau = 0.9 > \frac{1+\overline{\alpha_i}}{2}$ (0.89 in our experiments), which is consistent with our theoretical analysis in Section 5.4 that shows detection implies attribution in such settings. Our results show that watermark-based detection and attribution are accurate when the AI-generated, watermarked images are not post-processed. Specifically, the worst $TAR_i$ or $TDR_i$ is larger than 0.94; less than 0.1% of users have $TAR_i/TDR_i$ smaller than 0.98;

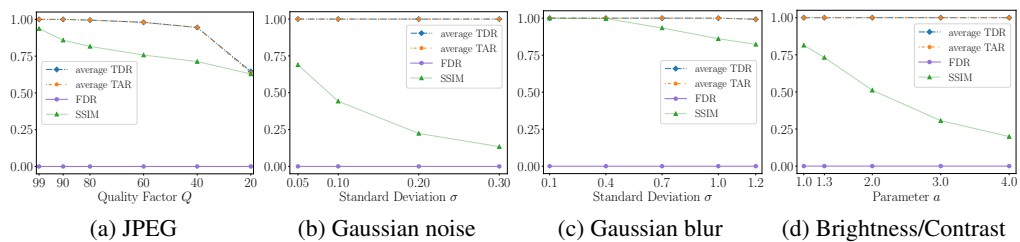

Figure 3: Detection and attribution results when AI-generated and non-AI-generated images are post-processed by common post-processing methods with different parameters. SSIM measures the quality of an image after post-processing.

and 85% of users have $TAR_i/TDR_i$ of 1 for Midjourney and DALL-E 2, and 60% of such users for Stable Diffusion.

**Impact of $s$, $n$, and $\tau$:** Figure 9 in Appendix shows the average *TDR*, average *TAR*, worst 1% *TDR*, worst 1% *TAR*, and *FDR* when $s$, $n$, or $\tau$ varies. Both average *TDR* and average *TAR* are close to 1, and *FDR* is close to 0, as $s$ varies from 10 to 1,000,000. The average *TDR* and average *TAR* slightly decrease when $n$ increases from 64 to 80, while the worst 1% *TDR/TAR* slightly increases as $n$ increases from 32 to 48 and then decreases as $n$ further increases. Our result implies that HiDDeN may be unable to accurately encode/decode very long watermarks. When $\tau$ increases, both average *TDR* and *TAR* decrease, while *FDR* also decreases. Such trade-off of $\tau$ is consistent with Theorem 1, 3, and 4.

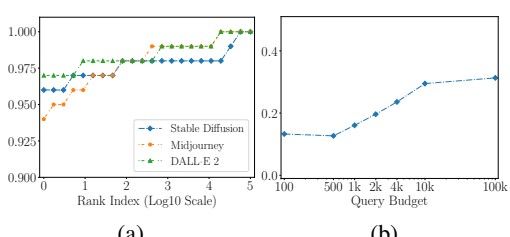

Figure 2: (a) Ranked $TAR_i$ of the 100,000 users. (b) Average SSIM between watermarked images and their adversarially post-processed versions as query budget varies in the black-box setting.

**Common post-processing:** Common post-processing is often used to evaluate the robustness of watermarking in *non-adversarial settings*. We use JPEG, Gaussian noise, Gaussian blur, and Brightness/Contrast, whose details are shown in Appendix J. We use adversarial training to train HiDDeN and the training details can be found in Appendix J. Figure 3 shows the detection/attribution results when a common post-processing method with different parameters is applied to the (AI-generated and non-AI-generated) images. Figure 3 also shows the average SSIM (Wang et al., 2004) between a (AI-generated and non-AI-generated) image and its post-processed version. Our results show that detection and attribution are robust to common post-processing. In particular, the average *TDR* and *TAR* are still high when a common post-processing does not sacrifice image quality substantially. For instance, average *TDR* and *TAR* start to decrease sharply when the quality factor $Q$ of JPEG is smaller than 40. However, the average SSIM between watermarked images and their post-processed versions also drops quickly. Figure 6 in Appendix shows a watermarked image and the versions post-processed by different methods.

**Adversarial post-processing:** Adversarial post-processing (Jiang et al., 2023) carefully perturbs a watermarked image to evade detection/attribution. HiDDeN is not robust to adversarial post-processing in white-box setting. Thus, HiDDeN-based detection/attribution is also not robust in such setting, i.e., *TDR/TAR* can be reduced to 0 while maintaining image quality.

Figure 2b shows the average SSIM between watermarked images and their adversarially post-processed versions in the black-box setting (i.e., WEvade-B-Q (Jiang et al., 2023)) as a function of the number of queries to the detection API for *each* watermarked image. Both *TDR* and *TAR* are 0 in these experiments since WEvade-B-Q always guarantees evasion (Jiang et al., 2023). However, adversarial post-processing substantially sacrifices image quality in the black-box setting (i.e., SSIM is small) even if an attacker can query the detection API for a large number of times. Figure 7 in Appendix shows several examples of adversarially post-processed images with degraded visual quality. Our results show that HiDDeN and thus our HiDDeN-based detection/attribution have good robustness to adversarial post-processing in the black-box setting.

### 6.3 COMPARING WATERMARK SELECTION METHODS

We compare three watermark selection methods: Random, NRG (Chen et al., 2016), and A-BSTA. NRG is the state-of-the-art approximate algorithm to the farthest string problem and we extend it to select watermarks (details in Appendix A). We do not use BSTA because it is not scalable, e.g., it takes more than 8 hours to select even 16 watermarks.

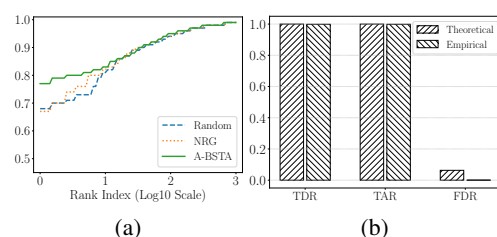

(a)                               (b)

Figure 4: (a) Ranked $TAR_i$ of the worst 1K users for the three watermark selection methods. (b) Theoretical vs. empirical results.

**Running time:** Table 3 in Appendix shows the running time to generate a watermark averaged among the 100,000 watermarks. Although A-BSTA is slower than Random and NRG, the running time is acceptable, i.e., it takes only 24ms to generate a watermark on average.

**TAR:** Figure 4a shows the ranked $TAR_i$ of the worst 1,000 users, where the AI-generated images are post-processed by JPEG compression with quality factor $Q = 90$ and HiDDeN is adversarially trained. The results indicate that A-BSTA outperforms NRG, which outperforms Random. This is because A-BSTA selects watermarks with smaller $\overline{\alpha_i}$, while Random selects watermarks with larger $\overline{\alpha_i}$ as shown in Figure 11 in Appendix.

### 6.4 THEORETICAL VS. EMPIRICAL RESULTS

We calculate the theoretical lower bounds of $TDR_i$ and $TAR_i$ of a user respectively using Theorem 1 and 4, while the theoretical upper bound of $FDR$ using Theorem 3. We estimate $\beta_i$ as the bitwise accuracy between the decoded watermark and $w_i$ averaged among the testing AI-generated

Table 1: Theoretical lower bounds of *TDR/TAR* and upper bound of *FDR* when there are 100 million users.

| Bound of TDR | Bound of FDR | Bound of TAR |
|---|---|---|
| 99.99% | 6.00% | 99.99% |

images, and estimate $\gamma$ using the fraction of bits in the decoded watermarks that are 1 among the non-AI-generated images. Figure 4b shows the average theoretical vs. empirical *TDR/TAR*, and theoretical vs. empirical *FDR*, when no post-processing is applied (Figure 12 in Appendix shows the results when JPEG with $Q = 90$ is applied). The results show that our theoretical lower bounds of *TDR* and *TAR* match with empirical results well, which indicates that our derived lower bounds are tight. The theoretical upper bound of *FDR* is notably higher than the empirical *FDR*. This is because some bits may have larger probabilities to be 1 or 0 in the experiments, but our theoretical analysis treats the bits equally, leading to a loose upper bound of *FDR*.

**Theoretical results when there are 100 millions users:** Due to limited computational resources, we show theoretical results on 100 million users in Table 1, assuming $\beta_i = 0.99$, $\underline{\alpha_i} = 0.2$, $\gamma = 0.05$, and $\overline{\alpha_i} = 0.8$. We notice that *TDR* and *TAR* remain very close to 1.

## 7 CONCLUSION AND FUTURE WORK

We show that watermark can be used for user-aware detection and attribution of AI-generated image. Moreover, via both theoretical analysis and empirical evaluation, we find that such detection and attribution inherit the accuracy/(non-)robustness properties of the watermarking method. We also find that selecting dissimilar watermarks for users enhances attribution performance.

**Text watermarking:** Our theory and algorithm may not be applicable to text watermarking (Kirchenbauer et al., 2023) that does not use bitstring as watermark, but is applicable to text watermarking (Abdelnabi & Fritz, 2021) that uses bitstring as watermark (Appendix K shows more details). Interesting future work is to extend our work to text or audio watermarking.

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

---

**Algorithm 1** BSTA($w_s, d, m$)

---

**Input:** Initial watermark $w_s$, recursion depth $d$, and $m$.
**Output:** $w_s$ or NotExist.

1: **if** $d < 0$ **then**
2:     return $NotExist$
3: **end if**
4: $i^* \leftarrow \arg\max_{i \in \{1,2,\cdots,s-1\}} BA(w_i, w_s)$
5: **if** $BA(w_{i^*}, w_s) > (m+d)/n$ **then**
6:     return $NotExist$
7: **else if** $BA(w_{i^*}, w_s) \leq m/n$ **then**
8:     return $w_s$
9: **end if**
10: $B \leftarrow \{k | w_s[k] = w_{i^*}[k], k = 1, 2, \cdots, n\}$
11: Choose any $B' \subset B$ with $|B'| = m+1$
12: **for all** $k \in B'$ **do**
13:     $w'_s \leftarrow w_s$
14:     $w'_s[k] \leftarrow \neg w'_s[k]$
15:     $w'_s \leftarrow \text{BSTA}(w'_s, d-1, m)$
16:     **if** $w'_s$ is not $NotExist$ **then**
17:         return $w'_s$
18:     **end if**
19: **end for**
20: return $NotExist$

---

## A WATERMARK SELECTION ALGORITHMS

**Bounded search tree algorithm (BSTA) (Gramm et al., 2003):** Recall that our watermark selection problem is equivalent to the farthest string problem. Thus, our decision problem is equivalent to that of the farthest string problem, which has been studied extensively in the theoretical computer science community. In particular, BSTA is the state-of-the-art *exact* algorithm to solve the decision problem version of the farthest string problem. We apply BSTA to solve the decision problem version of our watermark selection problem exactly, which is shown in Algorithm 1 in Appendix. The key idea of BSTA is to initialize $w_s$ as $\neg w_1$ (i.e., each bit of $w_1$ flips), and then reduce the decision problem to a simpler problem recursively until it is easily solvable or there does not exist a solution $w_s$. In particular, given an initial $w_s$, BSTA first finds the existing watermark $w_{i^*}$ that has the largest bitwise accuracy with $w_s$. If $BA(w_{i^*}, w_s) \leq m/n$, then $w_s$ is already a solution to the decision problem and thus BSTA returns $w_s$. Otherwise, BSTA chooses any $m+1$ bits that $w_s$ and $w_{i^*}$ match. For each of the chosen $m+1$ bits, BSTA flips the corresponding bit in $w_s$ and recursively solves the decision problem using the new $w_s$ as an initialization. The recursion is applied $m$ times at most, i.e., the recursion depth $d$ is set as $m$ when calling Algorithm 1.

A key limitation of BSTA is that it has an exponential time complexity Gramm et al. (2003). In fact, since the decision problem is NP-hard, all known *exact* solutions have exponential time complexity. Therefore, to enhance computation efficiency, we resort to approximate solutions. Next, we discuss the state-of-the-art approximate solution that adapts BSTA and a new approximate solution that we propose.

**Non Redundant Guess (NRG) (Chen et al., 2016):** Like BSTA, this approximate solution also first initializes $w_s$ as $\neg w_1$ and finds the existing watermark $w_{i^*}$ that has the largest bitwise accuracy with $w_s$. If $BA(w_{i^*}, w_s) \leq m/n$, then NRG returns $w_s$. Otherwise, NRG samples $n \cdot BA(w_{i^*}, w_s) - m$ bits that $w_s$ and $w_{i^*}$ match uniformly at random. Then, NRG flips these bits in $w_s$ and recursively solve the decision problem using the new $w_s$ as an initialization. Note that NRG stops the recursion when $m$ bits of the initial $w_s$ have been flipped. Algorithm 2 in Appendix shows NRG.

**Approximate bounded search tree algorithm (A-BSTA):** The algorithm of our A-BSTA is shown as Algorithm 3. Note that binary search is another way to find a proper $m$. Specifically, we start with a small $m$ (denoted as $m_l$) that does not produce a $w_s$ and a large $m$ (denoted as $m_u$) that does produce a $w_s$. If $m = (m_l + m_u)/2$ produces a $w_s$, we update $m_u = (m_l + m_u)/2$; otherwise

---

**Algorithm 2** $NRG(w_s, m)$

---

**Input:** Initial watermark $w_s$ and $m$.
**Output:** $w_s$ or NotExist.
1: $F \leftarrow \emptyset$
2: $d \leftarrow m$
3: **while** $d > 0$ **do**
4:      $i^* \leftarrow \arg\max_{i \in \{1,2,\cdots,s-1\}} BA(w_i, w_s)$
5:      **if** $BA(w_{i^*}, w_s) > 2m/n$ **then**
6:          return $NotExist$
7:      **else if** $BA(w_{i^*}, w_s) \leq m/n$ **then**
8:          return $w_s$
9:      **end if**
10:     $B \leftarrow \{k | w_s[k] = w_{i^*}[k] \wedge k \notin F, k = 1, 2, \cdots, n\}$
11:     $l \leftarrow n \cdot BA(w_{i^*}, w_s) - m$
12:     Sample $B' \subset B$ with $|B'| = l$ uniformly at random
13:     **for all** $k \in B'$ **do**
14:        $w_s[k] \leftarrow \neg w_s[k]$
15:     **end for**
16:     $d \leftarrow d - l$
17:     $F \leftarrow F \cup B'$
18: **end while**
19: return $NotExist$

---

**Algorithm 3** Solving our watermark selection problem

---

**Input:** Existing $s-1$ watermarks $w_1, w_2, \cdots, w_{s-1}$.
**Output:** Watermark $w_s$.
1: $m \leftarrow \max_{i \in \{1,2,\cdots,s-2\}} n \cdot BA(w_i, w_{s-1})$
2: **while** $w_s$ is $NotExist$ **do**
3:      **if** BSTA **then**
4:          $w_s \leftarrow \neg w_1$
5:          $w_s \leftarrow BSTA(w_s, m, m)$
6:      **end if**
7:      **if** NRG **then**
8:          $w_s \leftarrow \neg w_1$
9:          $w_s \leftarrow NRG(w_s, m)$
10:     **end if**
11:     **if** A-BSTA **then**
12:        $w_s \leftarrow$ sampled uniformly at random
13:        $w_s \leftarrow BSTA(w_s, d, m)$
14:     **end if**
15:     **if** $w_s$ is $NotExist$ **then**
16:        $m \leftarrow m + 1$
17:     **end if**
18: **end while**
19: return $w_s$

---

we update $m_l = (m_l + m_u)/2$. The search process stops when $m_l \geq m_u$. However, we found that increasing $m$ by 1 as in our Algorithm 3 is more efficient than binary search. This is because increasing $m$ by 1 expands the search space of $w_s$ substantially, which often leads to a valid $w_s$. On the contrary, binary search would require solving the decision problem multiple times with different $m$ until finding that $m + 1$ is enough.

**Time complexity:** We analyze the time complexity of the algorithms to solve the decision problem. For Random, the time complexity is $O(n)$. For BSTA, the time complexity to solve the decision problem with parameter $m$ is $O(snm^m)$ according to (Gramm et al., 2003). For NRG, the time complexity is $O(sn + s\sqrt{m} \cdot 5^m)$ according to (Chen et al., 2016). For A-BSTA, the time complexity is $O(snm^d)$, where $d$ is a constant.

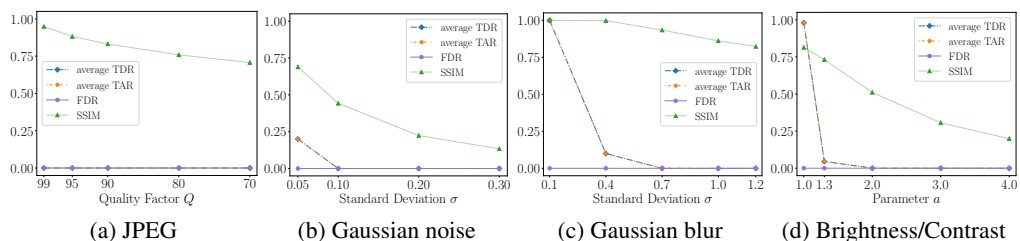

(a) JPEG  (b) Gaussian noise  (c) Gaussian blur  (d) Brightness/Contrast

Figure 5: Detection and attribution results when AI-generated and non-AI-generated images are post-processed by common post-processing methods with different parameters. HiDDeN is trained using standard training.

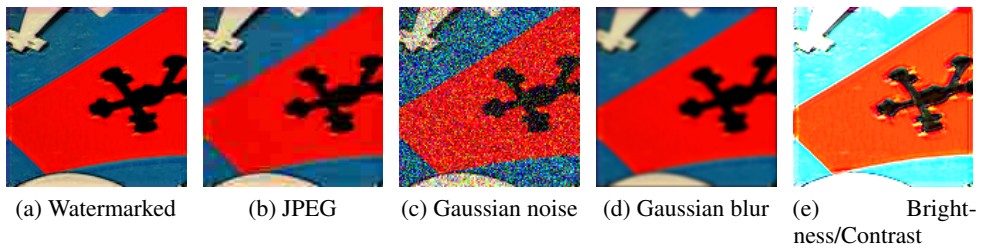

(a) Watermarked  (b) JPEG  (c) Gaussian noise  (d) Gaussian blur  (e) Brightness/Contrast

Figure 6: A watermarked image and the versions post-processed by JPEG with $Q$=20, Gaussian noise with $\sigma$=0.3, Gaussian blur with $\sigma$=1.2, and Brightness/Contrast with $a$=4.0.

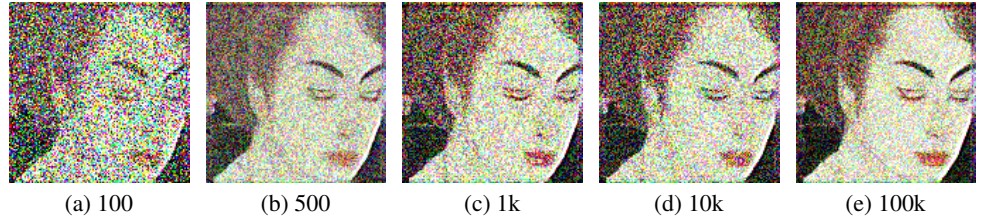

(a) 100  (b) 500  (c) 1k  (d) 10k  (e) 100k

Figure 7: Perturbed watermarked images obtained by adversarial post-processing with different number of queries to the detection API in the black-box setting.

## B    OTHER EVALUATION METRICS CAN BE DERIVED FROM $TDR_i$, $FDR$, AND $TAR_i$

We note that there are also other relevant detection and attribution metrics, e.g., the probability that an AI-generated image is incorrectly attributed to a user. We show that other relevant detection and attribution metrics can be derived from $TDR_i$, $FDR$, and $TAR_i$, and thus we focus on these three metrics in our work. Specifically, Figure 8 shows the taxonomy of detection and attribution results for non-AI-generated images and AI-generated images generated by user $U_i$. In the taxonomy trees, the first-level nodes represent ground-truth labels of images; the second-level nodes represent possible detection results; and the third-level nodes represent possible attribution results (note that attribution is only performed after an image is detected as AI-generated).

In the taxonomy trees, there are 5 branches in total, which are labeled as ①, ②, ③, ④, and ⑤ in the figure. Each branch starts from a root node and ends at a leaf node, and corresponds to a metric that may be of interest. For instance, our $TDR_i$ is the probability that an image $C \sim \mathcal{P}_i$ goes through branches ④ or ⑤; $FDR$ is the probability that an image $C \sim \mathcal{Q}$ goes through branch ②; and $TAR_i$ is the probability that an image $C \sim \mathcal{P}_i$ goes through branch ④. The probability that an image goes through other branches can be calculated using $TDR_i$, $FDR$, and/or $TAR_i$. For instance, the probability that a non-AI-generated image $C \sim \mathcal{Q}$ is correctly detected as non-AI-generated is the

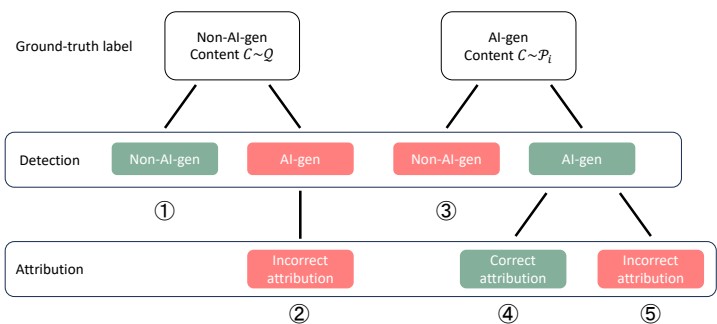

Figure 8: Taxonomy of detection and attribution results. Nodes with red color indicate incorrect detection/attribution.

probability that $C$ goes through the branch ①, which can be calculated as $1-FDR$. The probability that an AI-generated image $C \sim \mathcal{P}_i$ is incorrectly detected as non-AI-generated is the probability that $C$ goes through the branch ③, which can be calculated as $1-TDR_i$. The probability that a user $U_i$'s AI-generated image $C \sim \mathcal{P}_i$ is correctly detected as AI-generated but incorrectly attributed to a different user $U_j$ is the probability that $C$ goes through the branch ⑤, which can be calculated as $TDR_i-TAR_i$.

## C DEFINITIONS OF $\beta$-ACCURATE AND $\gamma$-RANDOM WATERMARKING

**Definition 3** ($\beta$-accurate watermarking). For a randomly sampled AI-generated image $C \sim \mathcal{P}$ embedded with the watermark $w$, the bits of the decoded watermark $D(C)$ are independent and each bit matches with that of $w$ with probability $\beta$, where $\beta \in [0, 1]$. Formally, we have $Pr(D(C)[k] = w[k]) = \beta$, where $C \sim \mathcal{P}$, $D$ is the decoder, and $[k]$ represents the $k$th bit of a watermark. We say a watermarking method is $\beta$-accurate if it satisfies the above condition.

**Definition 4** ($\gamma$-random watermarking). For a randomly sampled non-AI-generated image $C \sim \mathcal{Q}$ without any watermark embedded, the bits of the decoded watermark $D(C)$ are independent and each bit is 1 with probability at least $0.5 - \gamma$ and at most $0.5 + \gamma$, where $\gamma \in [0, 0.5]$. Formally, we have $|Pr(D(C)[k] = 1) - 0.5| \leq \gamma$, where $C \sim \mathcal{Q}$ and $[k]$ represents the $k$th bit of a watermark. We say a watermarking method is $\gamma$-random if it satisfies the above condition.

## D PROOF OF THEOREM 1

For $C \sim \mathcal{P}_i$, we denote $w = D(C)$, $n_i = BA(w, w_i)n$, and $n_j = BA(w, w_j)n$ for $j \in \{1, 2, \cdots, s\}/\{i\}$. Then we have the following:

$$|w - \neg w_i|_1 = n_i,$$
$$|\neg w_i - w_j|_1 = BA(w_i, w_j)n,$$
$$|w - w_j|_1 = n - n_j,$$

where $\neg w_i$ means flipping each bit of the watermark $w_i$, $| \cdot |_1$ is $\ell_1$ distance between two binary vectors. According to the triangle inequality, we have:

$$|w - w_j|_1 \leq |w - \neg w_i|_1 + |\neg w_i - w_j|_1$$
$$= n_i + BA(w_i, w_j)n.$$

Therefore, we derive the lower bound of $n_j$ for $j \in \{1, 2, \cdots, s\}/\{i\}$ as follows:

$$n_j = n - |w - w_j|_1$$
$$\geq n - n_i - BA(w_i, w_j)n.$$

Table 2: The maximum pairwise bitwise accuracy among the watermarks generated by NRG and A-BSTA for different initializations.

|  | $\neg w_1$ initialization | Random initialization |
|---|---|---|
| NRG | 0.766 | 0.750 |
| A-BSTA | 0.875 | 0.734 |

Table 3: The average running time for different watermark selection methods to generate a watermark.

|  | Random | NRG | A-BSTA |
|---|---|---|---|
| Time (ms) | 0.01 | 2.11 | 24.00 |

Thus, we derive the lower bound of *TDR$_i$* as follows:

$$TDR_i = 1 - \Pr(n_i < \tau n \wedge \max_{j \in \{1,2,\cdots,s\}/\{i\}} n_j < \tau n))$$

$$\geq 1 - \Pr(n_i < \tau n \wedge \max_{j \in \{1,2,\cdots,s\}/\{i\}} n - n_i - BA(w_i, w_j)n < \tau n))$$

$$= 1 - \Pr(n_i < \tau n \wedge n - n_i - \underline{\alpha_i} n < \tau n)$$

$$= 1 - \Pr(n - \tau n - \underline{\alpha_i} n < n_i < \tau n)$$

$$= \Pr(n_i \geq \tau n) + \Pr(n_i \leq n - \tau n - \underline{\alpha_i} n),$$

where $n_i \sim B(n, \beta_i)$ and $\underline{\alpha_i} = \min_{j \in \{1,2,\cdots,s\}/\{i\}} BA(w_i, w_j)$.

## E    PROOF OF COROLLARY 1

According to Theorem 1, the lower bound of *TDR$_i$* is $1 - \Pr(n - \tau n - \underline{\alpha_i} n < n_i < \tau n)$. For an integer $r \in (n - \tau n - \underline{\alpha_i} n, \tau n)$ and $n_i \sim B(n, \beta_i)$, we have the following:

$$\Pr(n_i = r) = \binom{n}{r} \beta_i^r (1 - \beta_i)^{n-r}.$$

Then we compute the partial derivative of the probability with respect to $\beta_i$ as follows:

$$\frac{\partial \Pr(n_i = r)}{\partial \beta_i} = \binom{n}{r} \beta_i^{r-1} (1 - \beta_i)^{n-r-1} (r(1 - \beta_i) - (n - r)\beta_i)$$

$$< \binom{n}{r} \beta_i^{r-1} (1 - \beta_i)^{n-r-1} (\tau - \beta_i) n.$$

The partial derivative is smaller than 0 when $\tau < \beta_i$. Therefore, the probability $\Pr(n_i = r)$ decreases as $\beta_i$ increases for any integer $r \in (n - \tau n - \underline{\alpha_i} n, \tau n)$. Thus, the lower bound of *TDR$_i$* increases as $\beta_i$ becomes closer to 1.

## F    PROOF OF THEOREM 2

For $C \sim Q$, we denote $n_1 = BA(D(C), w_1)n$ and $n_j = BA(D(C), w_j)n$ for $j \in \{1, 2, \cdots, s\}$. Then, we have the following:

$$FDR = 1 - \Pr(\max_{j \in \{1,2,\cdots,s\}} n_j < \tau n)$$

$$= 1 - \Pr(n_1 < \tau n \wedge \max_{j \in \{2,3,\cdots,s\}} n_j < \tau n).$$

To derive an upper bound of *FDR*, we denote:

$$|w - w_1|_1 = n - n_1,$$

$$|w_1 - w_j|_1 = n - BA(w_1, w_j)n,$$

$$|w - w_j|_1 = n - n_j.$$

According to the triangle inequality, we have the following:

$$|w - w_j|_1 \geq |w_1 - w_j|_1 - |w - w_1|_1$$

$$= n_1 - BA(w_1, w_j)n.$$

Therefore, we derive the upper bound of $n_j$ for $j \in \{2, 3, \cdots, s\}$ as follows:

$$n_j = n - |w - w_j|_1$$
$$\leq n - n_1 + BA(w_1, w_j)n.$$

Thus, we derive the upper bound of *FDR* as follows:

$$FDR = 1 - \Pr(n_1 < \tau n \wedge \max_{j \in \{2,3,\cdots,s\}} n_j < \tau n)$$
$$\leq 1 - \Pr(n_1 < \tau n \wedge \max_{j \in \{2,3,\cdots,s\}} n - n_1 + BA(w_1, w_j)n < \tau n))$$
$$= 1 - \Pr(n_1 < \tau n \wedge n - n_1 + \overline{\alpha_1}n < \tau n))$$
$$= 1 - \Pr(n - \tau n + \overline{\alpha_1}n < n_1 < \tau n)$$
$$= \Pr(n_1 \geq \tau n) + \Pr(n_1 \leq n - \tau n + \overline{\alpha_1}n),$$

where $n_1 \sim B(n, 0.5)$ and $\overline{\alpha_1} = \max_{j \in \{2,3,\cdots,s\}} BA(w_1, w_j)$.

## G  PROOF OF THEOREM 3

For $C \sim Q$, we denote $n_j = BA(D(C), w_j)n$ for $j \in \{1, 2, \cdots, s\}$, and we have the following:

$$FDR = 1 - \Pr(\max_{j \in \{1,2,\cdots,s\}} n_j < \tau n)$$
$$= 1 - \prod_{j \in \{1,2,\cdots,s\}} \Pr(n_j < \tau n).$$

According to Definition 4, for any $k \in \{1, 2, \cdots, n\}$ and any $j \in \{1, 2, \cdots, s\}$, the decoding of each bit is independent and the probability that $D(C)[k]$ matches with $w_j[k]$ is at most $0.5 + \gamma$ no matter $w_j[k]$ is 1 or 0. Therefore, we have the following:

$$FDR = 1 - \prod_{j \in \{1,2,\cdots,s\}} \Pr(n_j < \tau n)$$
$$\leq 1 - \Pr(n' < \tau n)^s,$$

where $n'$ follows the binomial distribution with parameters $n$ and $0.5 + \gamma$, i.e., $n' \sim B(n, 0.5 + \gamma)$.

## H  PROOF OF COROLLARY 2

According to Theorem 3, the probability $Pr(n' < \tau n)$ increases when $\gamma$ decreases. Therefore, the upper bound of *FDR* decreases as $\gamma$ becomes closer to 0.

## I  PROOF OF THEOREM 4

For $C \sim \mathcal{P}_i$, we denote $w = D(C)$, $n_i = BA(w, w_i)n$, and $n_j = BA(w, w_j)n$ for $j \in \{1, 2, \cdots, s\}$. Then we have the following:

$$|w - \neg w_i|_1 = n_i,$$
$$|\neg w_i - w_j|_1 = BA(w_i, w_j)n,$$
$$|w - w_j|_1 = n - n_j.$$

According to the triangle inequality, we have:

$$|w - w_j|_1 \geq |w - \neg w_i|_1 - |\neg w_i - w_j|_1$$
$$= n_i - BA(w_i, w_j)n.$$

Therefore, we derive the upper bound of $n_j$ for $j \in \{1, 2, \cdots, s\}/\{i\}$ as follows:

$$n_j = n - |w - w_j|_1$$
$$\leq n - n_i + BA(w_i, w_j)n.$$

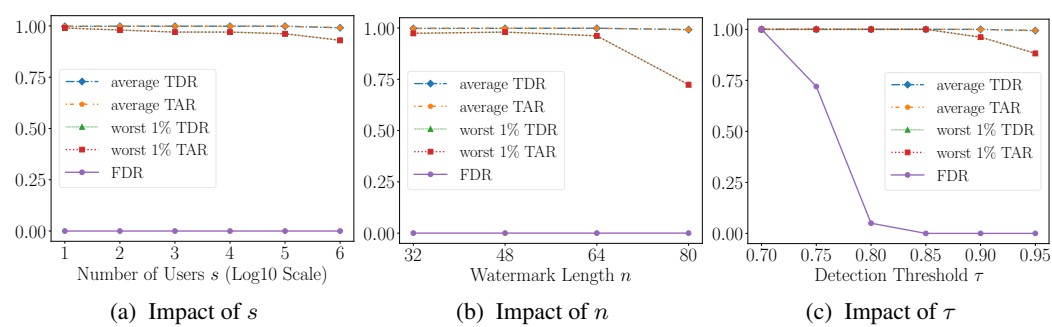

(a) Impact of $s$       (b) Impact of $n$       (c) Impact of $\tau$

Figure 9: Impact of number of users $s$, watermark length $n$, and detection threshold $\tau$ on detection and attribution performance.

Thus, we derive the lower bound of $TAR_i$ as follows:

$$
\begin{aligned}
TAR_i =& \Pr(\max_{j\in\{1,2,\cdots,s\}} n_j \geq \tau n \wedge n_i > \max_{j\in\{1,2,\cdots,s\}/\{i\}} n_j)\\
\geq& \Pr(\max_{j\in\{1,2,\cdots,s\}} n_j \geq \tau n \wedge n_i > \max_{j\in\{1,2,\cdots,s\}/\{i\}} n - n_i + BA(w_i, w_j)n)\\
=& \Pr(\max_{j\in\{1,2,\cdots,s\}} n_j \geq \tau n \wedge n_i > \frac{n + \overline{\alpha_i}n}{2})\\
=& \Pr(\max_{j\in\{1,2,\cdots,s\}} n_j \geq \tau n \wedge n_i > \frac{n + \overline{\alpha_i}n}{2} \mid n_i \geq \tau n) \cdot \Pr(n_i \geq \tau n)\\
&+ \Pr(\max_{j\in\{1,2,\cdots,s\}} n_j \geq \tau n \wedge n_i > \frac{n + \overline{\alpha_i}n}{2} \mid n_i < \tau n) \cdot \Pr(n_i < \tau n)\\
\geq& \Pr(n_i > \frac{n + \overline{\alpha_i}n}{2} \mid n_i \geq \tau n) \cdot \Pr(n_i \geq \tau n)\\
=& \Pr(n_i > \frac{n + \overline{\alpha_i}n}{2} \wedge n_i \geq \tau n)\\
=& \Pr(n_i \geq \max\{\lfloor \frac{1 + \overline{\alpha_i}}{2}n \rfloor + 1, \tau n\}),
\end{aligned}
$$

where $n_i \sim B(n, \beta_i)$ and $\overline{\alpha_i} = \max_{j\in\{1,2,\cdots,s\}/\{i\}} BA(w_i, w_j)$.

## J  COMMON POST-PROCESSING AND ADVERSARIAL TRAINING

**Common post-processing:** Each of these post-processing methods has some parameters, which control the size of perturbation added to a (watermarked or unwatermarked) image.

**JPEG.** JPEG method (Zhang et al., 2020) compresses an image via a discrete cosine transform. The perturbation introduced to an image is determined by the *quality factor Q*. An image is perturbed more when $Q$ is smaller.

**Gaussian noise.** This method perturbs an image via adding a random Gaussian noise to each pixel. In our experiments, the mean of the Gaussian distribution is 0. The perturbation introduced to an image is determined by the parameter *standard deviation $\sigma$*.

**Gaussian blur.** This method blurs an image via a Gaussian function. In our experiments, we fix kernel size $s = 5$. The perturbation introduced to an image is determined by the parameter *standard deviation $\sigma$*.

**Brightness/Contrast.** This method perturbs an image via adjusting the brightness and contrast. Formally, the method has contrast parameter $a$ and brightness parameter $b$, where each pixel $x$ is converted to $ax + b$. In our experiments, we fix $b = 0.2$ and vary $a$ to control the perturbation.

**Adversarial training (Zhu et al., 2018):** We use adversarial training to train HiDDeN. Specifically, during training, we randomly sample a post-processing method from no post-processing and common post-processing with a random parameter to post-process each watermarked image in a mini-batch.

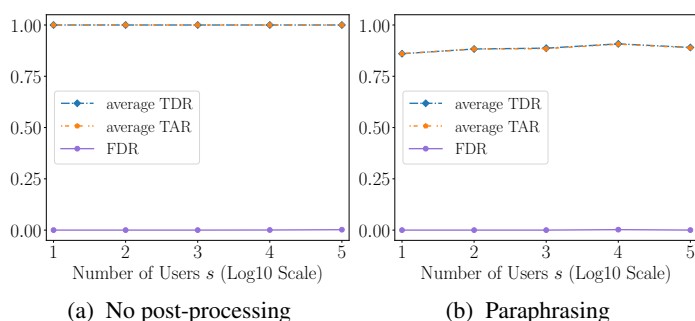

(a) No post-processing  (b) Paraphrasing

Figure 10: Results of watermark-based detection and attribution for AI-generated texts.

Following previous work (Zhu et al., 2018), we consider the following range of parameters during adversarial training: $Q \in [10, 99]$ for JPEG, $\sigma \in [0, 0.5]$ for Gaussian noise, $\sigma \in [0, 1.5]$ for Gaussian blur, and $a \in [1, 20]$ for Brightness/Contrast.

## K  DETECTION AND ATTRIBUTION OF AI-GENERATED TEXTS

Our method can also be used for the detection and attribution of AI-generated texts based on text watermarking that uses bitstring as watermark. For text watermarking, we use a learning-based method called *Adversarial Watermarking Transformer (AWT)* (Abdelnabi & Fritz, 2021). Given a text, AWT encoder embeds a bitstring watermark into it; and given a (watermarked or unwatermarked) text, AWT decoder decodes a watermark from it. Following the original paper (Abdelnabi & Fritz, 2021), we train AWT on the word-level WikiText-2 dataset, which is derived from Wikipedia articles (Merity et al., 2017). We use most of the hyperparameter settings in the publicly available code of AWT except the weight of the watermark decoding loss. To optimize watermark decoding accuracy, we increase this weight during training. The detailed hyperparameter settings for training can be found in Table 4.

We use A-BSTA to select users' watermarks. For each user, we sample 10 text segments from the test corpus uniformly at random, and perform watermark-based detection and attribution. Moreover, we use the unwatermarked test corpus to calculate FDR. Figure 10 shows the detection and attribution results when there is no post-processing and *paraphrasing* (Damodaran, 2021) is applied to texts, where $n = 64$, $\tau = 0.85$, and $s$ ranges from 10 to 100,000. Due to the fixed-length nature of AWT's input, we constrain the output length of the paraphraser to a certain range. When paraphrasing is used, we extend adversarial training to train AWT. Specifically, we employ T5-based paraphraser to post-process the watermarked texts generated by AWT. Due to the non-differentiable nature of the paraphrasing process, we cannot jointly adversarially train the encoder and decoder since the gradients cannot back-propagate to the encoder. To address the challenge, we first use the standard training to train AWT encoder and decoder. Then, we use the encoder to generate watermarked texts, paraphrase them, and use the paraphrased watermarked texts to fine-tune the decoder. The detail parameter settings of fine-tuning are shown in Table 4.

Note that the average *TDR/TAR* and *FDR* are all nearly 0 when AWT is trained by standard training and paraphrasing is applied to texts. The results show that our method is also applicable for AI-generated texts, and adversarially trained AWT has better robustness to paraphrasing.

## L  ATTRIBUTION OF GENAI SERVICES

In this work, we focus on attribution of the image to users for a specific GenAI service. Another relevant attribution problem is to trace back the GenAI service (e.g., Google's Imagen, OpenAI's DALL-E 3, or Stable Diffusion) that generated a given image. Our method can also be applied to such GenAI-service-attribution problem by assigning a different watermark to each GenAI service. When GenAI service generates an image, its corresponding watermark is embedded into it. Then, our method can be applied to detect whether an image is AI-generated and further attribute the GenAI service if the image is detected as AI-generated.

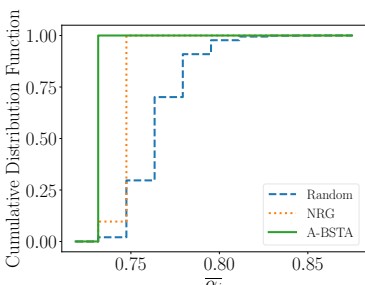

Figure 11: The cumulative distribution function (CDF) of $\overline{\alpha_i}$.

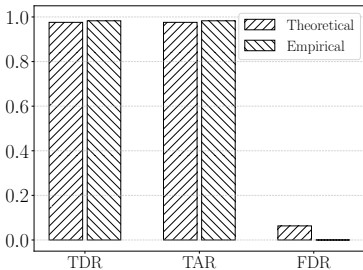

Figure 12: Theoretical vs. empirical results when JPEG with $Q = 90$ is applied.

Table 4: Default parameter settings for the training of AWT.

| Phase | Standard Training | Fine-Tuning |
|---|---|---|
| Optimizer | Adam | |
| # epochs | 200 | 10 |
| Batch size | 16 | |
| Learning rate | $3 \times 10^{-5}$ | |
| # warm-up iterations | 6000 | 1000 |
| Length of text | 250 | $250 \pm 16$ |
| Generation weight | 1.5 | 1 |
| Message weight | 10000 | |
| Reconstruction weight | 1.5 | 2 |

In service attribution, selecting watermarks for different GenAI services can be coordinated by a central authority, who runs our watermark selection algorithm to pick unique watermarks for the GenAI services. A GenAI service registers to the central authority in order to obtain a unique watermark. Such a central authority is similar to the certificate authority in the Public Key Infrastructure (PKI) that is widely used to secure communications on the Internet. The central authority may also perform detection and attribution of AI-generated images since it has access to all GenAI services' watermarks. However, the central authority may become a bottleneck in such detection and attribution. To mitigate this issue, the watermarks of all GenAI services can be shared with each GenAI service, so each GenAI service can perform detection and attribution. We note that a central authority is not needed in user attribution of a particular GenAI service. This is because the GenAI service can select watermarks for its users and perform detection/attribution.

**Hierarchical attribution:** We can perform attribution to GenAI service and user simultaneously. Specifically, we can divide the watermark space into multiple subspaces; and each GenAI service uses a subspace of watermarks and assigns watermarks in its subspace to its users. In this way, we can trace back both the GenAI service and its user that generated a given image.

