# OpenReview forum: "Watermark-based Detection and Attribution of AI-Generated Image"
_ICLR.cc/2025/Conference — ICLR 2025 Conference Withdrawn Submission_

### Official Review · Reviewer_M2WU · 2024-10-31

**Soundness:** 2
**Presentation:** 2
**Contribution:** 2
**Rating:** 5
**Confidence:** 4

**Summary:**

In this paper, the authors study watermark (bit string) selection problem. They aim to increase to maximize the minimum distance between watermarks and use bounded search tree algorithms to solve this maximization problem. Then, they present the definition of TDR, FDR and TAR and give their theoretical low bounds. Experiments are also conducted.

**Strengths:**

1)	The authors give clear definitions of their terms.
2)	In addition to provide algorithm and experimental results, the authors also give some theoretical explanations.

**Weaknesses:**

1)	Bit string selection problem has been well studied and its solution has been applied to different domains. The literature review should cover these works.

2)	The watermark selection problem is related to error correction code methods. The literature review should cover them, and comparison is expected.

3)	Some other works have discussed the statistics of watermarks and their detection. The authors should say clearly the difference between their work and the previous one, e.g.,

    (a)	Watermark detection algorithm using statistical decision theory

    (b)	https://openreview.net/pdf?id=Fc2FaS9mYJ

4)	The authors should do more extensive survey to make sure that this problem has not been done by watermarking (images, video, LLM, and many others) and information theory communities.

5)	The proposal method is a straightforward application of an existing mathematical tools. No novelty is from the algorithm.

6)	The authors compute the low bounds of TDR, FDR and TAR. I believe that the authors can compute the exact probability. It is not clear to me why they compute the lower bound.

7) As long as the bit length is large enough, this assignment problem should be less much less important than enhancing the robustness of the detector. The watermarked images can be suffered from various different editing and modification.

**Questions:**

See the weaknesses

---

### Official Review · Reviewer_5FFr · 2024-11-02

**Soundness:** 3
**Presentation:** 3
**Contribution:** 1
**Rating:** 3
**Confidence:** 3

**Summary:**

The paper explores watermark-based methods for detecting and attributing AI-generated images, aiming to identify the user responsible for generating an image. The authors develop theoretical bounds for detection and attribution performance, provide an algorithm to select unique watermarks, and test their framework against both adversarial and common post-processing techniques. The paper claims improvements in user-aware detection, watermark selection, and empirical testing of detection robustness under various conditions.

**Strengths:**

- The paper addresses the issue of user attribution for AI-generated content, a field that has seen limited exploration.

- The authors provide rigorous probabilistic analysis for detection and attribution, contributing to the theoretical grounding of watermarking methods.

- Comprehensive testing across multiple AI models (e.g., Stable Diffusion, Midjourney) with various post-processing techniques highlights the robustness of the proposed method.

- The use of an adapted farthest string problem algorithm demonstrates an efficient way to assign watermarks, enhancing attribution reliability.

**Weaknesses:**

- The problem of assigning unique watermarks could be approached more straightforwardly by using a simpler scheme, such as naively assigning binary keys from 0 to $2^k-1$ to the users and encoding these with an Error Correction Code (ECC), to expand them into $n$-bit watermark keys. This would allow efficient error detection and correction without complex algorithms, questioning the necessity of the algorithm developed in the paper. The authors have not mentioned ECC methods and have not compared their algorithms to those in practice, which is a necessity in my opinion. I suggest the authors compare their algorithm against existing ECCs (e.g. polar codes, BCH), in terms of fundamental and theoretical differences and similarities, and practical results such as watermark detection accuracy.
For reference, I suggest authors investigate works such as [1] that leverage ECC for watermarking.

- The paper bases its experiments on HiDDeN, a deep-learning watermarking model, but this may not generalize to other watermarking methods (e.g., StegaStamp [2], Trustmark [3], TreeRing [4]) as different watermarking techniques have different properties and robustness issues against different types of attacks. A broader evaluation of different watermarking techniques would enhance the applicability of the findings. For weaknesses and strengths in the robustness of some popular existing watermarking methods, I refer authors to evaluation papers such as [5] [6].

- For a more comprehensive evaluation, it would be necessary to include more attacks (post-processing) in Section 6.2 (refer to [5] for a comprehensive list of attacks, or some papers proposing watermarking techniques, such as [3] [4], for a smaller set of common attacks).

- I believe one of the most important contributions of the paper is shown in Figure 4(a), where the performance of the proposed algorithm is compared to the random and NRG methods. I believe the paper needs more focus and more experimental results for this setting, presenting the significant statistical improvement of their method compared to naive approaches and previous work. I ask the authors to correct me if I'm wrong, but my understanding from Figure 4(a) is that A-BSTA provides no improvement over the random selection, except for the bottom 10 users (out of 100k users), which seems a minimal improvement.

[1] Saberi, M., Sadasivan, V.S., Zarei, A., Mahdavifar, H. and Feizi, S., 2024. DREW: Towards Robust Data Provenance by Leveraging Error-Controlled Watermarking.

[2] Tancik, M., Mildenhall, B. and Ng, R., 2020. Stegastamp: Invisible hyperlinks in physical photographs.

[3] Bui, T., Agarwal, S. and Collomosse, J., 2023. TrustMark: Universal Watermarking for Arbitrary Resolution Images.

[4] Wen, Y., Kirchenbauer, J., Geiping, J. and Goldstein, T., 2023. Tree-ring watermarks: Fingerprints for diffusion images that are invisible and robust

[5] An, B., Ding, M., Rabbani, T., Agrawal, A., Xu, Y., Deng, C., Zhu, S., Mohamed, A., Wen, Y., Goldstein, T. and Huang, F., 2024. Benchmarking the robustness of image watermarks.

[6] Saberi, M., Sadasivan, V.S., Rezaei, K., Kumar, A., Chegini, A., Wang, W. and Feizi, S., 2023. Robustness of ai-image detectors: Fundamental limits and practical attacks.

**Questions:**

- As explained in weaknesses, what are the similarities and differences between the proposed method and ECCs?

- Considering Figure 4(a), how do the authors claim that the improvements of their method compared to random selection are significant? What is the improvement in the overall accuracy of the detection (over all users)?

- What if instead of selecting $2^k$ $n$-bit user keys, we use a watermarking method with $k$-bit keys, and select the user keys naively? Have the authors performed any experiments and comparisons for this scenario?

---

### Official Review · Reviewer_8vqX · 2024-11-03

**Soundness:** 2
**Presentation:** 1
**Contribution:** 2
**Rating:** 5
**Confidence:** 3

**Summary:**

This paper introduces a study on watermark-based detection and attribution of AI-generated images. The innovation involves a theoretical analysis of detection and attribution performance, as well as the development of an algorithm for selecting watermark bitstrings.

**Strengths:**

The paper presents upper and lower bound analyses for the three metrics proposed by the authors, focusing on detection and attribution performance. In addition, the paper also analyzes the complexity of selecting watermark bitstrings and presents an efficient approximation algorithm.

**Weaknesses:**

1. The experiments in the paper rely solely on one watermarking method, demonstrating the effectiveness of the proposed method. It is recommended to conduct experiments with other watermarking methods for a comprehensive comparison.
2. The watermark selection algorithm only compares with one metric, TAR, which is incomprehensiveness in demonstrating the effectiveness of A-BSTA. The authors are advised to evaluate the proposed method with other two metrics, TDR and FDR.
3. The adaption on the three metrics porposed in this paper are not obvious compared with previous works. A detailed explanation on why these metrics are designed in this way would be appreciated. In addition, the theoretical analysis provided does not offer guidance for the experiments.
4. In terms of the quality of the paper, there is room for improvement. The writing feels sloppy and in many parts it was quite hard to follow the logic and understand the actual information. (e.g., "Generative AI (GenAI) can synthesize very realistic-looking images" lacks academic rigor.)

**Questions:**

1. The paper employs a long-running deterministic algorithm that is then trimmed into an approximation algorithm. Was there any consideration of directly using an approximation algorithm instead?

---

### Official Review · Reviewer_jEjX · 2024-11-04

**Soundness:** 3
**Presentation:** 2
**Contribution:** 2
**Rating:** 5
**Confidence:** 3

**Summary:**

This paper explores watermark-based detection and attribution methods for AI-generated images, aiming to identify not only if an image is AI-generated but also the specific user who created it. The authors apply watermarking techniques by embedding unique watermarks for each registered user in a generative AI service. They adapt existing algorithms to improve watermark selection efficiency, which helps reduce the likelihood of false attributions among users.

**Strengths:**

This paper provides a structured approach to explore watermark-based attribution.
Robustness Against Common Post-Processing: The watermark demonstrates reasonable resilience against typical post-processing transformations, such as JPEG compression and Gaussian blur, which are frequently encountered in image pipelines.

**Weaknesses:**

The paper assumes that the proposed detection and attribution framework can generalize across different watermarking methods. However, the experiments demonstrate the method’s effectiveness using only a single type of watermarking technique

**Questions:**

Same as weakness

---

### Note · Authors · 2024-11-13

I have read and agree with the venue's withdrawal policy on behalf of myself and my co-authors.